# The Role of Tripartite Motif Family Proteins in Chronic Liver Diseases: Molecular Mechanisms and Therapeutic Potential

**DOI:** 10.3390/biom14081038

**Published:** 2024-08-21

**Authors:** Xiwen Cao, Yinni Chen, Yuanli Chen, Meixiu Jiang

**Affiliations:** 1The Queen Mary School, Jiangxi Medical College, Nanchang University, 999 Xuefu Road, Nanchang 330031, China; jp4217120192@qmul.ac.uk; 2The National Engineering Research Center for Bioengineering Drugs and the Technologies, Institute of Translational Medicine, Jiangxi Medical College, Nanchang University, 999 Xuefu Road, Nanchang 330031, China; 407400220014@email.ncu.edu.cn; 3Key Laboratory of Major Metabolic Diseases, Nutritional Regulation of Anhui Department of Education, College of Food and Biological Engineering, Hefei University of Technology, Hefei 230002, China; chenyuanli@hfut.edu.cn

**Keywords:** CLDs, TRIM proteins, MAFLD, HBV, HCV, mechanisms, clinical perspective

## Abstract

The worldwide impact of liver diseases is increasing steadily, with a consistent upswing evidenced in incidence and mortality rates. Chronic liver diseases (CLDs) refer to the liver function’s progressive deterioration exceeding six months, which includes abnormal clotting factors, detoxification failure, and hepatic cholestasis. The most common etiologies of CLDs are mainly composed of chronic viral hepatitis, MAFLD/MASH, alcoholic liver disease, and genetic factors, which induce inflammation and harm to the liver, ultimately resulting in cirrhosis, the irreversible final stage of CLDs. The latest research has shown that tripartite motif family proteins (TRIMs) function as E3 ligases, which participate in the progression of CLDs by regulating gene and protein expression levels through post-translational modification. In this review, our objective is to clarify the molecular mechanisms and potential therapeutic targets of TRIMs in CLDs and provide insights for therapy guidelines and future research.

## 1. Introduction

Chronic liver diseases (CLDs) are a degenerative condition that deteriorates liver function over a period of more than six months. It affects an excess of 300 million people around the world and caused 1.32 million deaths in 2017 [1]. It is characterized by an ongoing process of inflammation, liver tissue destruction, and regeneration, which ultimately results in cirrhosis and fibrosis [2]. The main CLDs discussed in this review include chronic viral hepatitis, MAFLD/MASH, and liver fibrosis during CLD development.

Hepatitis B virus (HBV) and hepatitis C virus (HCV) tend to be the prevailing occurrence in patients with viral hepatitis. The affliction caused by HBV represents a grave medical challenge on a global scale. HBV is usually typically characterized by HBsAg, which can persistently be detected exceeding 6 months [3]. Individuals afflicted with long-term HBV infections face the danger of progressing to end-stage liver disease, including cirrhosis and hepatocellular carcinoma (HCC). Hepatitis C is caused by HCV, with a 1.6% global prevalence. Approximately 50–90% of patients who suffer from acute infection become chronically infected [4]. The advancement of HCV can result in liver fibrosis, cirrhosis, HCC, and mortality. chronic hepatitis is mainly transmitted via blood, vertical transmission, and intimate encounters involving intercourse. After entering human circulation, the virus mainly infects hepatocytes and triggers a series of inflammatory reactions [5].

Nowadays, metabolic dysfunction-associated steatotic liver disease (MAFLD) is one of the second most prevalent CLDs worldwide [6]. In contrast to the old name “NAFLD”, the diagnosis of MAFLD also involves metabolic risk factors. The latest study indicates that NAFLD findings remain applicable under the MASLD definition [7]. Over the past eight years, the prevalence in the global adult population has increased from 25% to 38% [8]. This etiology arises from a complex interplay of diet, exercise, genetic factors, body weight, and metabolism. Metabolic dysfunction-associated steatohepatitis (MASH) is the most severe mode of MAFLD. With chronic liver disease progression, it inevitably gives rise to the development of liver fibrosis [9].

Liver fibrosis is a persistent abnormal regenerative reaction triggered by a variety of long-term tissue damage, which include the widespread accumulation of ECM and abnormal proliferation of connective tissue [10]. As a complex pathological process, liver fibrosis is a reversible pathological progression caused by CLDs, which may finally lead to cirrhosis, liver failure, or even HCC if left untreated. In developed countries, the prime causes of liver fibrosis are chronic hepatitis, alcohol, and metabolic syndrome [11].

In the field of therapeutics, antiviral therapy is a recognized and effective treatment that is widely used for viral hepatitis. However, there is limited access to diagnosis and treatment. The other CLDs mentioned above lack targeted treatment. Although liver transplantation is considered a radical treatment, the problem of rejection and related complications still needs to be solved. Current treatment is mainly used to relieve symptoms and delay disease progression by taking indirect drugs and improving living habits. Therefore, new therapeutic targets need to be developed.

TRIM proteins typically work as E3 ubiquitin ligases, which participate in various biological processes and control crucial cellular functions through promoting protein ubiquitination and degradation [12]. After reviewing the number of articles, we found that the expression level of TRIM fluctuates when CLDs occur, and they are important to the interaction of various proteins in CLD-related mechanisms. The special modification of TRIM proteins makes them a promising diagnostic and therapeutic target. As a potential diagnostic biomarker, the detection of the TRIM protein is of great significance; in addition to helping to evaluate the progression and prognosis of liver disease, it can also be used to detect drug therapy effect and advance preventive screening [13]. Except for TRIM proteins, other markers such as ADAPT and MACK-3 have also been shown to evaluate the progression of liver fibrosis and MASH. This also improves the efficiency of screening people for clinical drug trials, avoiding invasive tests such as liver biopsies [14]. Once we discover the complete cellular pathway they regulate and their specific role in it, we can manipulate cellular function by regulating the expression of TRIM proteins to achieve effective therapeutic outcomes. Protease inhibitors have been demonstrated as an effective adjunctive treatment for liver-related diseases. With the advancement of precision medicine, there is no doubt that this topic holds significant research value [15].

In order to comprehensively review and analyze the advanced mechanism of TRIM proteins in CLDs, a systematic search strategy of the literature was adopted in this review. We mainly focused on relevant research results in the past few years. Searches were made in multiple academic databases, including PubMed, Web of Science, Scopus, and Google Scholar. The search terms included “TRIM protein”, “HBV”, “HCV”, “MAFLD (NAFLD)”, ”MASH(NASH)”, and “chronic liver disease”, which were combined with the study type and language category. Through this process, we aim to collect and analyze the most representative and influential studies in the literature within this field to provide a comprehensive perspective. Based on these studies, this review summarizes TRIM proteins’ function and discusses the underlying mechanisms, which may prove TRIM proteins to be an innovation tool for the therapeutic potential of CLDs.

## 2. TRIM Family

### 2.1. Structure of TRIM Proteins

TRIM proteins belong to the RING-E3 ligases and have a number of members exceeding 80 in the human genome [16]. The definition involves an N-terminal TRIM motif comprising the RING finger domain, one or two b-box zinc finger domains, and the coiled-coil domain [17]. Together, these fragments comprise the TRIM family-specific RBCC domain. The RING finger domain helps to promote interactions between proteins for enzymatic activity. The B-box domain consists of concise peptide sequences and is significant in identifying target proteins [18,19]. The coiled-coil domain is situated following the B-box within TRIM proteins and contributes to assembling both homomeric and heteromeric complexes [20].

Typically, the RBCC domain is highly conserved so that the distinct variations in the TRIM structure are determined by various additional C-terminal domains [21]. According to the diversity of the C-terminal domain, TRIM proteins are distributed to 11 subgroups from C-I to XI. It is worth noting that there is also a group of isoforms in the TRIM family called the UC group, which refers to those who lack RING domains, but some of them still function as E3 ligases [22] (Figure 1). Distinct C-terminal structures possess diverse functions. Of these, PRY/SPRY, also known as B30.2, is a common domain in the C-terminal site that identifies and binds target molecules for regulating innate immune responses and host–pathogen interactions [23,24,25]. Others include the COS, FN3, ARF, ACID, BROMO, FIL, NHL, PHD, MATH, and TM domain. More commonly, the COS domain takes part in dimerization and binds to the microtubule cytoskeleton [26]. The FN3 domain interacts with DNA; the PHD domain regulates transcription by mediating histone binding [23,27]. The MATH domain mediates transcription factor function by interacting with the TNF receptor [28]. These abundant C-terminal domains allow them to bind to different substrates and play different regulatory roles, which determines the specificity and functional diversity of TRIM [12].

### 2.2. Function of TRIM Proteins

Proteins contain a RING-finger domain with the E3 ubiquitin ligase comprising most TRIM family proteins. This means they are able to be involved with the ubiquitin–proteasome pathway to modify protein degradation [29].

Ubiquitin is a 76-residue highly conserved protein in eukaryotic cells [16]. The ubiquitin–proteasome system (UPS) consists of ubiquitin (Ub); three types of ubiquitinates include E1 (Ub activating enzymes), E2 (Ub conjugating enzymes), E3 (Ub ligases), deubiquitinases (DUBs) and the proteasome (Figure 2), which is crucial in the process for post-translational modification, which is widely discussed in the scientific community [30,31]. E3 ligases constitute most members of UPS, and they can select target proteins and then link one or more types of ubiquitin to lysine residues of the protein substrate [32]. This is an ATP-dependent enzymatic cascade-regulating protein for homeostasis and degradation. Moreover, studies have shown that ubiquitination participates in various other forms of regulation, such as protein trafficking, activated signaling pathway transduction, and chromatin remodeling.

In general, lysine-48 (K-48) and lysine-63 (K-63) represent the predominant types of ubiquitination observed in mammalian species. The modification of the K-48 linkage mainly involves targeting proteins for proteasomal degradation, while K-63 linkage modification is used for coupling and mediating nonproteolytic signals, including regulating the signal of subcellular localization, protein activation, and protein interaction [33]. The C-IV subgroup, as the largest TRIM subgroup that produces all major ubiquitin modifications, is also the most well-studied subgroup [34]. TRIM proteins are crucial in numerous biological processes due to their participation in the ubiquitination process. It also affects cell activities, which makes the TRIM protein an important player in numerous diseases, including functional disorders, viral infections, and cancer [35].

## 3. TRIM Proteins in CLD

### 3.1. TRIM and Hepatitis B Virus Infection

The hepatitis B virus insidiously establishes itself during the initial phase of infection. Upon entering the host cell, the virus genome moves into the nucleus and transforms into cccDNA, serving as the blueprint for transcription. This process is mediated and intervened by adaptive immune response [36]. When CD8+T cells are activated and migrate into the liver, they exhibit antigen recognition capabilities and effectively eliminate infected cells [5,37]. They also secrete IFN-γ, triggering an extensive cascade to amplify the inflammatory process. However, HBV may adopt an active escape strategy against adaptive immune response, resulting in a decrease in CD8+T cell reactivity. This allows HBV replication at high levels in the patient’s liver. The inefficient immune response leads to hepatocyte destruction and long-term regeneration.

The HBV has 3.2 kb genomic DNA and produces four distinct mRNAs to encode the surface (HBs), pre-core (HBe), core (HBc), polymerase (pol), and x (HBx) antigens [38]. TRIM proteins interact with the mRNA and their products to regulate the survival and proliferation of viruses together (Figure 3).

#### 3.1.1. TRIM Proteins Interact with HBx

HBx promotes genome replication through the cccDNA template [39]. The most recent study indicates that the promotion of MAN1B1 expression by the HBx protein is achieved through the TRIM25-mediated enhancement of GRP78 stability, thereby inhibiting its ubiquitination [40]. This could partially explain the transformation mechanism from HBV patients to HCC. TRIM14 is a key regulator in the IFN-related pathway, which induces IL-27 secretion and directly activates STAT1 [41]. It also acts as a STAT1-dependent ISG and interacts with HBx to inhibit the SMC-HBx-DDB1 complex formation. Simultaneously, TRIM14 strongly hinders the breakdown of HBx on Smc5/6, preventing HBx from facilitating HBV replication [42]. The TRIM5γ gene hinders virus replication by facilitating K48 ubiquitination to destroy HBx at the K95 site in the BBox domain [43]. TRIM5γ can also recruit TRIM31 for binding and degrading HBx.

#### 3.1.2. TRIM Proteins Interact with HBV Pol

HBV DNA Pol consists of three functional domains (RT, TP, and RNaseH) and a variable spacer region [44]. In order to make HBV DNA less stable, the SPRY domain of TRIM21 negatively acts on the TP at Lys260 and Lys283 sites [45]. It inhibits HBV DNA replication by mediating the K48 ubiquitin–proteasome pathway to promote HBV Pol DNA degradation.

#### 3.1.3. TRIM Proteins Interact with HBc

The presence of TRIM26 is essential for the replication of HBV in cells that have been infected to avoid proteasomal-dependent HBc degradation in a concentration-dependent manner [46]. Similar to TRIM14, TRIM25 functions as an ISG and is induced by IFN-α signaling via the IL-27 activation of STAT1 and STAT3 [47]. It inhibits HBV replication by preventing HBV core promoter (CP) activity, which is produced by pgRNA and is able to regulate the viral life cycle. Research shows that TRIM25 mediates the ubiquitination and activation of RIG-I, which is a viral sensor [47]. TRIM22 can also restrain the activity of HBV CP in an analogous manner. Meanwhile, TRIM22 is significantly elevated after IFN-γ treatment [48]. IFN-γ also induces the TRIM56 level by stimulating the JAK/STAT signaling pathway, while it enhances the phosphorylation of p65, which is a molecule in the NF-κB signaling pathway required for the HBV replication inhibitor by ubiquitinating IκBα through the RING domain [49]. Activated p65 inhibits HBV core promoter activity and HBV CP activity during cccDNA transcription to impair HBV amplification [50].

### 3.2. TRIM and Hepatitis C Virus Infection

HCV disrupts the homeostasis of the liver microenvironment, leading to oxidative stress and immune responses in virus-infected hepatocytes, resulting in inflammation reactions. In addition, infected hepatocytes also produce multiple growth factors, chemokines, and cytokines, which participate in the recruitment of immune cells and local inflammatory response. Furthermore, fibrogenesis is a major complication of chronic HCV infection [51].

RNA in HCV is translated into a precursor polyprotein, then cleaved into 10 distinct proteins. E1, E2, and the core protein are three structural proteins. The remaining seven nonstructural proteins include p7, NS2, NS3, NS4A, NS4B, NS5A, and NS5B [52]. These host proteins interconnect and co-form the HCV replicase complex.

Studies show that some of the TRIM family members play a key role in the change in viral load expression in HCV patients. TRIM14 degrades the NS5A1 subdomain through K48-linked ubiquitination by binding NS5A [53]. NS5A is a viral component that facilitates the establishment of HCV in host cells by inhibiting the signaling pathway of IFN-α [54]. The overexpression of TRIM14 has been proven to effectively inhibit HCV infection and replication in hepatocytes. TRIM22 inhibits HCV replication by ubiquitinating NS5A [55]. The alteration in early viral kinetics suggests that TRIM22 engages in IFN-α induced antiviral effects. The specific interaction of the SPRY domain and D1 is required between TRIM22 and NS5A.

Through CRISPR-Cas9 genome evaluation, TRIM26 has been identified as an important factor in facilitating HCV replication [56]. It facilitates K27-linked ubiquitination at residue K51 by interacting with the NS5B protein encoded by HCV, which promotes NS5B and NS5A binding. TRIM26 has also taken part in HCV species tropism. TRIM25 was found to be an antiviral protein, which significantly increased in untreated HCV patients [57]. However, the mechanism of HCV still needs to be explored.

### 3.3. TRIM and MAFLD/MASH

Depending on the severity and acute degree of disease progression, MAFLD can be categorized as steatosis, steatohepatitis, and, ultimately, cirrhosis. The detailed pathogenesis of MAFLD is still unclear. The two-hit theory may explain the progression of MAFLD. Initially, the accumulation of free fatty acids causes disturbances in lipid metabolism. Then, it results in a second hit, including insulin resistance (IR), inflammation, and oxidative stress, ultimately leading to liver damage, steatosis, and fibrosis. The intrahepatic inflammatory response is crucial for the occurrence of MAFLD, in which the abnormal activation and infiltration of Kuffer cells (liver macrophages) lead to liver injury. MAFLD can be divided into two forms: the initial type is associated with metabolic syndrome, which identifies IR as the main mechanism, while the second form typically arises from viral infection and medication usage [58].

Some TRIM family proteins take part in the progress of MAFLD (Figure 4). Compared with normal liver tissues, MAFLD tissues exhibit an increased expression of TRIM59 and are proven to interact with GPX4, a molecule that reduces the availability of lipid peroxides and stimulates ferroapoptosis through glutathione (GSH) to promote its ubiquitination. Furthermore, they utilized a specific inhibitor of ferroptosis, deferoxamine (DFO), and found that DFO inhibits TRIM59-induced cellular steatosis, which indicates that ferroapoptosis is important in MAFLD [59]. However, our previous studies have shown that LPS downregulated TRIM59 expression by regulating Sp1 and Nrf1, and the overexpression of TRIM59 can inhibit macrophage inflammation [60,61]. Moreover, Zheng Jin. et al. also found that TRIM59 can inhibit inflammation and protect the liver from damage by promoting phagocytosis [62]. Therefore, the role of TRIM59 in the liver microenvironment may be inconsistent and needs further investigation. TRIM65 expression is prevented by LPS through the ERK1/2 signaling pathway in macrophages and plays an important role as a negative regulator controlling the NLRP3 inflammasome to inhibit inflammatory response [19,63], which may assist in inhibiting the development of MAFLD. However, the function of TRIM65 in hepatocytes needs further investigation.

MAFLD progression heavily relies on the crucial role played by the TAK1-TAB2/3-MAPK pathway. After inhibiting the TAK1-JNK pathway, TNIP3 (TNFAIP3-interacting protein 3) mitigates MASH. It can inhibit hepatic lipid accumulation and downregulate MASH. TNIP3 interacts directly via the K158 ubiquitination site of TAK1, a kinase protein of MAPK kinase. It additionally impedes the signaling cascade by interacting with the AHD1 domain to promote TAK1 K63 ubiquitination. The overexpression of TRIM8 leads to insulin resistance, hepatic steatosis, inflammation, and fibrosis. TRIM8 directly promotes TAK1 phosphorylation and activates downstream pathways, including c-Jun-p38 and NK-kB signaling [64]. TRIM16 is significantly upregulated in lipotoxicity and attenuated MAFLD progression on HFD and HFHC diets. TRIM16 improves MAFLD by degrading phosphorylated TAK1, which can inhibit the JNK-p38 signal transduction pathway. Multi-omics analysis indicates that the progression of MASH is impeded by TRIM16 through suppressing the activation of the MAPK signaling pathway [65]. TRIM38 also ameliorates the progression of MAFLD by negatively regulating MAPK. It also interacts with TAK1 binding protein 2 (TAB2) to promote its degradation [66].

TRIM21 is downregulated in patients with MAFLD and directly interacts with fatty acid synthase (PEPCK1) and FASN and degrades them through the RING domain in hepatocytes to rectify hepatic glucose and lipid metabolism [67]. TRIM24 was found to inhibit spontaneous hepatic lipid accumulation in mice, and the knockout mice developed hepatocellular, steatosis, and HCC. Genome-wide RNA expression analysis showed how TRIM24 participates in biological functions such as lipid synthesis, storage, metabolism, and VLDL transporter expression. Furthermore, TRIM24 is also enriched in genes linked to inflammation and liver injury [68]. CEBPD is a pro-inflammatory factor that induces liver inflammation, glucose and lipid metabolism, insulin resistance, and liver fibrosis [69]. TRIM26 was found to inhibit CEBPD-HIF1A signaling and their cascade molecules, such as NF-κB, p38, and p65, by interacting with and degrading it, thereby delaying the progression of MASH [70]. According to reports, the key MAFLD repressor scaffold protein sorting nexin 8 (SNX8) recruits TRIM28, which then promotes FASN degradation through K48 ubiquitination [71], which indicates that TRIM28 may have a potential inhibitory effect on the development of MAFLD.

TRIM31 inhibits Rhbdf2-MAP3K7 and its downstream signaling by directly binding alongside polyubiquitination-degrading Rhbdf2 to prevent hepatic steatosis [72]. TRIM67 expression was reported to be induced by PGC-1α, HFD, and obesity, which activates liver inflammation, the accumulation of liver lipids, and the progression of MAFLD. TRIM6, TRIM9, TRIM22, and TRIM69 were significantly upregulated in tissues with severe MAFLD [59]. However, their functions and specific molecular mechanisms still remain to be studied.

### 3.4. TRIM and Liver Fibrosis

Hepatic fibrosis is marked by the activation of hepatic stellate cells (HSCs) and the overproduction of collagen, which is an inflammatory and fibrosis process [73]. A large number of inflammatory mediators activate HSCs, subsequently promoting their differentiation into myofibroblasts after liver injury. The stimulated HSCs continuously secrete the extracellular matrix, which is abundant in collagen I and III, and finally causes collagen accumulation and fibrotic scar formation, leading to fibrosis [10].

TRIM proteins play a vital part in regulating fibrosis progress. TRIM26 is downregulated in fibrosis tissues, and it interacts with SLC7A11, which can facilitate glutamate release, the uptake of cystine, and generate GSH to eliminate lipid ROS by promoting its breakdown via ubiquitination [74]. That is to say, TRIM26 can alleviate CCl4-induced liver fibrosis in mice models by promoting lipid peroxidation via inhibiting the SLC7A11-mediating process to increase Fe^2+^ release and the downregulation of GSH and NAPDH, thereby activating ferroapoptosis to inhibit HSCs activation [75].

TRIM33, also called TIF1γ, is a negative regulator in fibrosis through the TGF-β pathway. TRIM33 is expressed in the Disse space of the normal liver and is reduced in liver fibrosis. It blocks TGF-β1 signaling by competitively binding to SMAD4 and activating SMAD2/3. The binding of TRIM33 to SMAD2/3 on the α-smooth muscle actin (αSMA) promoter is enhanced by HGF, a growth factor in the liver, resulting in decreased αSMA expression. HGF also promotes the binding between pCREBs133 and the CREB site of the TRIM33 promoter to enhance its expression level [76]. TRIM52 promotes fibrosis by enhancing the activation of HSC by downregulating the accumulation of PPM1A (an anti-fibrotic molecule) to mediate the Smad 2/3 pathway. Moreover, TRIM52 may promote the development of fibrosis through the PPM1A-mediated TGF-β/Smad pathway in vitro [77]. TRIM37 facilitates liver fibrosis by interacting with SMAD7 and promoting ubiquitin degradation. This mechanism enhances TGF-β activity by inhibiting the TGF-β negative feedback loop mediated by Smad7. Meanwhile, the HBV-mediated transcriptional activation of NF-κB promotes TRIM37 expression and, in turn, its expression is induced by HBx protein-produced ROS, which are located upstream of TRIM37 and also promotes the transcriptional activity of NF-κB and aggravates liver fibrosis [78]. TRIM23 expression exhibits a positive correlation with liver fibrosis. Moreover, the augmented expression of TRIM23 enhances HSC activation by inhibiting ferroapoptosis. Additionally, it facilitates p53 ubiquitination, resulting in the attenuation of the SLC7A11 expression and the elevation of SAT1 and GLS2 expression for the promotion of ferroapoptosis [79,80].

Knocking down TRIM15 suppressed the activation of HSCs, which indicates that it enhances the proliferation and migration of HSCs. It positively regulates αSMA expression and type I collagen. Moreover, TRIM15 recruits focal adhesion kinase (FAK), which activates HSC activity to enhance fibrosis [81]. This finding infers that the FAK signaling pathway may be involved in HSC physiological activity. TRIM8 and TRIM24 have been proven to enhance gene expression that promotes fibrosis in fatty liver mice models [64,68]. Patients who are coinfected with HCV and HIV are in danger of liver fibrosis linked to TRIM5 and TRIM22 [82]. However, the involved mechanisms remain unknown.

So, to sum up, TRIM proteins mainly play an E3 ligase function in CLDs and regulate cellular pathways to affect disease progression by targeting protein degradation, as shown in Table 1 and Table 2.

## 4. Clinical Perspective

Even today, using medicine to treat liver diseases by regulating TRIM proteins has not been commercially popularized. Although drugs targeting the TRIM protein are still under development, many clinical trials have shown that the expression level of the TRIM protein directly affects curative effect and prognosis. With the development of precision medicine, more novel therapies can be used for related treatments.

### 4.1. TRIM in HBV

Up to now, two main classes of drugs have been recommended for the treatment of chronic hepatitis, including NAs and Peg-IFNα. Because of their negative response in part of the population, the evaluation and optimization of treatment regimens are becoming the focus of research [83].

Using the TRIM protein as a biomarker of HBV may become a future research direction. The change in expression levels of the TRIM protein is significant in the clinical prediction of virological responses. The regulation of TRIM19, 38, and 25 is related to HBsAg clearance and peg-IFN-α treatment response [40,84]. TRIM38 plays an antiviral role as an ISG, which is stimulated by IFN-α to inhibit HBV replication and upregulates Mx1, IFIT1, and STAT1 simultaneously [84].

Previous studies have also reported that the use of recombinant IFN-γ for therapy in diverse cell types can increase the mRNA level of TRIM26. Based on a genetic database study, patients carrying the CC/CT genotype at the rs116806878 locus exhibited a significantly elevated rate of HBeAg seroconversion, indicating that rs 116806878 can be used as a biomarker for predicting the effect of PegIFN-α treatment in CHB patients [85]. Exploring more SNP functions could build a powerful and complete prediction system for various diseases.

### 4.2. TRIM in HCV

As an inhibitor of the NS5B polymerase in HCV, sofosbuvir functions as an NA and is commonly prescribed alongside other DAAs for all genotypes of hepatitis [86]. In addition, pegIFN-α therapy inhibits viral replication by constantly stimulating the expression of ISG in the infiltrating immune cells of the liver. DAAs are more frequently used as a result of lower adverse reactions and higher sustained viral response (SVR) rates [87].

TRIM14 is elevated in RVR (rapid virological response) patients compared with SVR patients [53]. In clinical research, the TRIM22 mRNA expression level was significantly elevated in individuals exhibiting RVR and early virological response (EVR) [57]. The TRIM22 mRNA expression level is correlated with peg-IFNα-2a/ribavirin combination treatment and can be used to evaluate the effectiveness of treatment outcomes. PBMC extracted from SVR patients have been detected to have higher TRIM22 mRNA levels than relapse patients. Sustained viral clearance, along with high TRIM22 expression, was observed in more than 90% of patients treated with DAAs [88]. The TRIM22 expression level could also be correlated with the prevention of relapse in follow-up studies.

As personalized medicine becomes more popular and GWAS databases keep growing, some SNPs in TRIM protein genes that are encoded by ISGs may affect how often people recover after being diagnosed with HCV. Patients carrying TRIM5 rs 3824949 (GG), TRIM22 rs 7113258 (AA), and rs 1063303 (GC) are more likely to obtain RVR, EVR, and SVR after pegIFN-α/RBV treatment [89,90]. SNP genotyping may help recognize patients who require antiviral therapy in the early stage. This allows doctors to save medical resources and improve treatment outcomes.

### 4.3. TRIM in MAFLD/MASH

The increasing incidence of MAFLD/MASH will become the main reason for liver transplantation in the next decade, whereas there is no drug for a radical cure at present. Improving exercise and dietary habits, as well as reducing body weight, can slow the progression of the disease to a certain extent. Liver transplantation is a radical treatment for cirrhosis, but it usually faces severe rejection reactions. Some potential relieving drugs are used, such as hypoglycemic drugs, statins, lipid-lowering drugs, and antioxidants [9]. The intake of vitamin E, herbal compounds, and the Mediterranean diet was shown to be beneficial in patients with MAFLD. Liraglutide, currently used as a first-line treatment for T2DM, has been found to reduce liver inflammation by regulating macrophage polarization through the cAMP-PKA/STAT3 signaling cascade [91]. It also alleviated hepatic steatosis by activating the IRS2/PI3K/Akt signaling cascade [92]. Five phase III studies of the new drug (aramchol, resmetiron, obeticholic acid, belapectin, lanifibranor, efruxifermin) are looking to be completed by the end of 2024 [9].

Studies show how introducing TNIP3 using adenovirus in the liver effectively halts the advancement of MASH in mice [93]. TRIM8, located downstream of TNIP3, could be a potential therapeutic target of MASH. Moreover, the potent lipogenic capacity of lipase makes it a promising target. Previous clinical trials have demonstrated how TRIM21 reduces de novo lipogenesis and steatosis in hepatocytes by inhibiting FASN; this mechanism is detected in patients with TRIM21, which reduces hepatic de novo lipogenesis and steatosis in patients with obesity and MAFLD and obesity [67]. Targeting the SNX8-FASN axis to increase the TRIM28 expression level is a promising method for MAFLD treatment [71]. Targeting TRIM16 to interact with the MAPK pathway axis is also a promising method of MAFLD therapy, which is worthy of further study in clinical practice. Targeting TRIM26 to improve MASH by inhibiting the CEBPD-HIF1A signaling pathway can also be considered a potentially critical target. Furthermore, research shows that TRIM proteins are believed to modulate the NLRP3 inflammasome, thereby controlling inflammatory reactions and mitigating oxidative stress [22]. Targeted studies on TRIM proteins and related signaling pathways provide a basis for the future treatment and prevention of MAFLD.

### 4.4. TRIM in Liver Fibrosis

At present, most drugs for liver fibrosis are still in the stages of clinical trials. Methods like reducing inflammation and oxidation, stopping the growth of HSCs, and speeding up ECM degradation have been used to inhibit the growth of liver fibrosis.

With this in-depth study, increasingly therapeutic targets, pathway mechanisms, and emerging technologies have been discovered. These provide innovative ideas for liver fibrosis therapy. Ubiquitination and regulatory functions make TRIM proteins become promising therapeutic targets. Maintaining TRIM33 and TRIM26 levels can become a potential therapeutic approach during liver injury [75]. TRIM33 response to hE-MSC therapy is a promising therapeutic option that can reduce liver surface fluctuation and the fibrosis area without immune response [76]. In regenerative medicine, hE-MSCs are the ideal origin in stem cells. Targeting TRIM37 to regulate the TGFβ-1 signaling pathway in HSCs is promising for treatment [78]. Enhancing PPM1A expression by targeting TRIM52 is a promising strategy to block liver fibrosis [77]. Aldrovandi et al. found that extreme iron accumulation treated by acetaminophen in hepatocytes leads to liver fibrosis in a mouse model [94,95]. Therefore, TRIM23-mediated ferroapoptosis is used as a promising therapeutic for ameliorating fibrosis by targeting HSCs. Researchers successfully used erastin and sorafenib to alleviate liver fibrosis in mice models by modulating autophagy and inducing ferroapoptosis [96]. With the development of gene therapy, siRNA and microRNA (miRNA)-based treatment is promising.

## 5. Conclusions

In this essay, we overview the research progress of the function and mechanisms of TRIM family members, which belong to the E3 ubiquitin ligases in CLDs, and provide insights for future research and treatment discovery for CLDs.

TRIM proteins have a conserved N-terminal RBCC domain and a diverse C-terminal domain, and their specific structure makes them able to regulate ubiquitin proteins. TRIM proteins regulate HBV development at both gene and protein levels. HBx and HBc were found to be the most modified targets of TRIM. Interestingly, TRIM not only promoted ubiquitination in this process but also inhibited it. It also regulates transcription and replication in IFN-α-induced ISG production. There are relatively few studies on TRIM in HCV compared to others, but there are clinical trials supporting the evaluation of efficacy and prognosis. One thing is certain: the NS5A protein is the most important component structure of HCV; it interacts with a variety of TRIM proteins; and protease inhibitors have also been proven to assist in the conventional treatment of HCV, which is promising. While TRIM proteins play a role in regulating viral replication, the viral load also affects the expression level of the protein. For example, TRIM22 upregulation is associated with HCV decline during IFN-α treatment. At the same time, in some TRIM proteins that play a role in ISG (TRIM14, TRIM25, etc.), the level of the virus indirectly promotes the expression of TRIM proteins [55].

The mechanism of TRIM on MAFLD can be mainly divided into two categories. One is the regulation of signaling molecules in cellular pathways, which is ultimately reflected in NF-κB intracellular and extracellular pathway molecules so as to maintain or destroy hepatic lipid homeostasis. The other is to directly or indirectly regulate the activity of FASN by controlling this key rate-limiting enzyme for lipid synthesis. This suggests that they could be promising therapeutic targets and points the way for future research. As a systemic-effect metabolic disease, MAFLD is closely associated with IR and extrahepatic metabolic diseases. Hepatic insulin resistance has been found to be regulated by the TRIM protein, and we can speculate that the expression of the TRIM protein is widespread and may play an important role in the metabolic pathways of various organs [97]. Hepatic fibrosis, regarding the progression of MAFLD and hepatitis, has some mechanisms in common with them, including lipid peroxidation. The process of ferroapoptosis is controlled by targeting GSH and cell membrane receptors. At the same time, TRIM proteins also regulate the activation of the TGF-β/Smad signaling pathway, which is the core mechanism of HSC activation. The proliferation and migration ability of HSC is also affected by the level of TRIM expression.

To sum up, CLDs are a continuous development process, and TRIM proteins show a series of concentrated, correlated, and multidimensional regulatory functions in their role, which undoubtedly provides the exact direction for CLD exploration (Figure 5). With the development of precision medicine, some alleles encoding TRIM have been shown to be specific in the development and treatment of diseases. Treatment at the gene level may become the key point of future research.

At the same time, we also found that the progression of CLDs is continuous, and different liver diseases can be transformed into each other; patients with hepatitis B and C are prone to HCC, and liver fibrosis is the progressive form of MAFLD/MASH. TRIM proteins play an important regulatory role in these diseases. Previous studies have shown that the expression level of TRIM proteins changes significantly during the progression of HBV infection to HCC, and TRIM proteins also regulate the lipid peroxidation mechanism involved in both MAFLD and liver fibrosis [98]. This strong correlation can provide us with new ideas for exploring targets.

Although in recent years, the studies and achievements of TRIM proteins related to CLDs and HCC have been extremely popular, in the research on TRIM proteins and liver disease, there are also some problems to be solved. First of all, the function and structure of some TRIM family members are still unclear, and some specific domains need to be further studied. Many members have a similar structure and function of redundancy; their specific interactions with other proteins need to be discovered. The expression of TRIM proteins is universal and diverse. Even from the same subfamily with a similar structure, it may also have uncertain effects. Some can play the same role in the face of the same disease, such as TRIM28 and TRIM38 in the progression of MAFLD and TRIM15 and TRIM47 in liver fibrosis. There are also opposite regulatory effects, such as TRIM21 and TRIM26 in HBV infection and TRIM11, TRIM22, and TRIM26 in HCV infection. Surprisingly, the same molecule can play multiple roles in the same disease, such as TRIM25, which inhibits viral replication while also promoting HCC progression. This means that in order to ensure a curative effect, the future treatment of TRIM needs to rely on precision medicine therapy to accurately target drug delivery.

In recent years, proteolysis-targeting chimeras (PROTACs) technology has become an updated target for protein degradation. PROTAC is a bifunctional molecule capable of concurrently binding to E3 ligase and is a target protein for proteasomal recognition and degradation [99]. Unfortunately, at present, this technology is only used in anti-tumor therapy. Based on the principles of PROTAC research and investigations, we suggest that the treatment of this UPS could be spread to CLDs, and target proteins prone to aggregation could be selected to control the degradation of the UPS system function. However, it is necessary to possess a thorough, complete function of TRIM proteins in liver disease, which poses challenges for future studies. At the same time, some TRIM proteins may have a bidirectional effect on the disease, which does not promote or inhibit the progression of the disease alone but does so through complex and multiple mechanisms, which means that we need to conduct more detailed studies. So, constructing a complete biological system of TRIM proteins will undoubtedly help to develop targeted drugs and provide innovative ideas for further research.

## Figures and Tables

**Figure 1 biomolecules-14-01038-f001:**
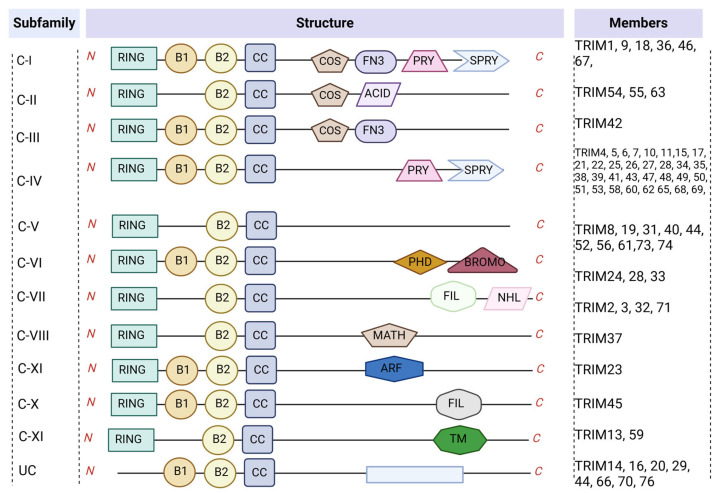
The classification of TRIM proteins based on structure. TRIM proteins are divided into 11 groups depending on variable C-terminal domains. RING finger domain, B1, B2-box zinc finger domains, and coiled-coil domain located at the N-terminal constitute a conservative RBCC structure domain. PRY/SPRY, COS, FN3, ACID, FIL, NHL, PHD, BROMO, ARF, MATH, and TM domains are close to the C-terminal. They determine the species of TRIM proteins.

**Figure 2 biomolecules-14-01038-f002:**
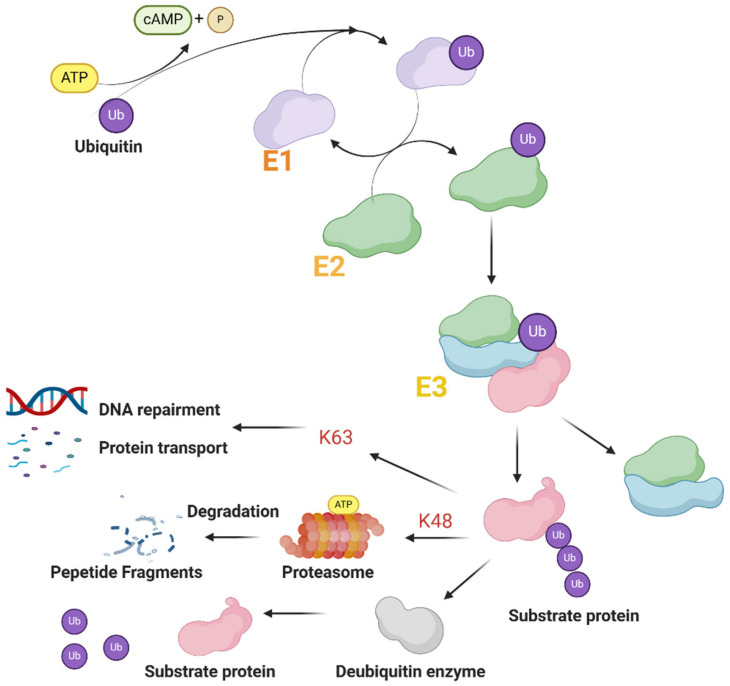
The ubiquitin–proteasome degradation pathway of proteins. E1, E2, and E3 groups are activated sequentially. E2 and E3 form a complex to bind the substrate protein and ubiquitinate it, which is degraded by the proteasome into small molecule peptide chains; this process can be reversed by DUB.

**Figure 3 biomolecules-14-01038-f003:**
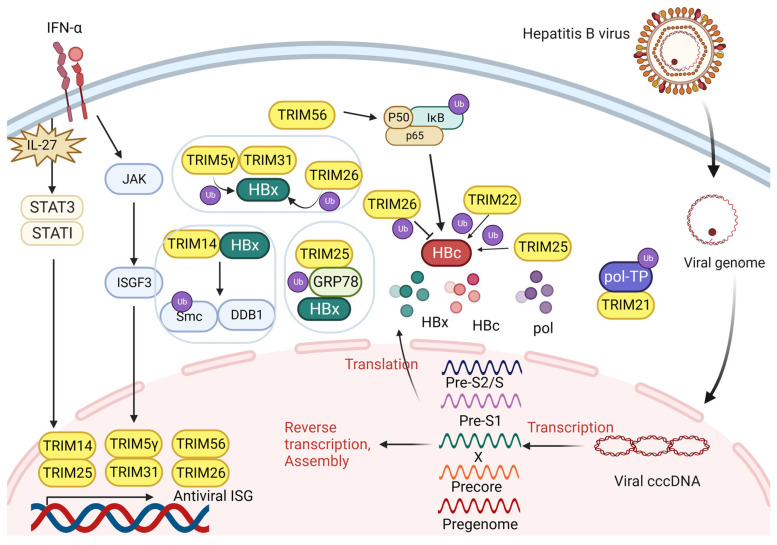
Mechanisms of interaction between TRIM proteins and HBV-related proteins. After HBV enters the nucleus and transcribes in the form of cccDNA, different mRNAs are produced to encode the surface (HBs), pre-nuclear (HBe), core (HBc), HBV polymerase (pol), and x (HBx) antigens. By interacting with proteins, TRIM proteins directly or indirectly affect viral replication. After patients are stimulated by IFN-a treatment, some TRIM proteins are activated for transcription as ISGs and participate in regulatory processes.

**Figure 4 biomolecules-14-01038-f004:**
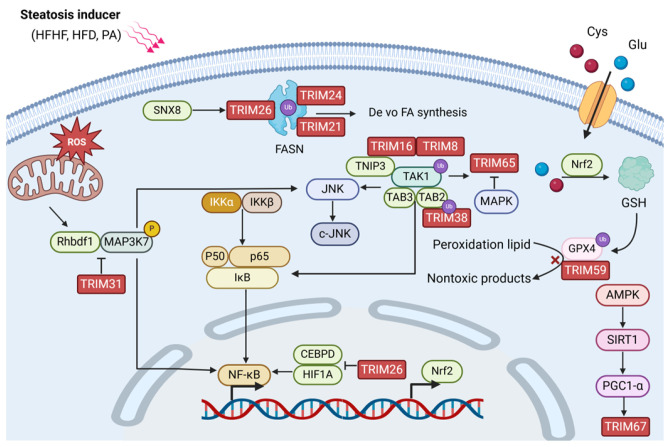
TRIM proteins in the cellular signaling pathway of MAFLD. The figure shows the mechanism of MAFLD, mainly involving NF-κB, MAPK, AMPK, and other signaling pathways. TRIM proteins regulate the disease by targeting its upstream and downstream molecules at both the gene and protein levels. MAFLD is also regulated by combining FASN, which is a key enzyme in lipid synthesis, the lipid peroxide clearance process, and ferroapoptosis.

**Figure 5 biomolecules-14-01038-f005:**
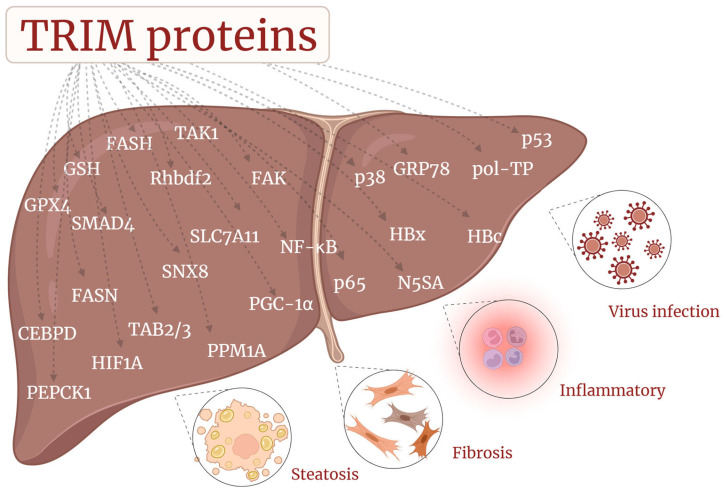
Summary of target molecules in the CLDs modulating the mechanism of TRIM proteins. The figure summarizes the molecules associated with the major regulatory mechanisms covered in the article, corresponding to the related diseases they regulate (viral hepatitis, MASLD/MASH, etc.).

**Table 1 biomolecules-14-01038-t001:** The function and mechanism of the TRIM protein promoting the progression of CLDs.

Diseases	TRIM Members	Functions	Molecular Mechanisms
HBV	TRIM26	Single-nucleotide polymorphisms (SNPs) of rs116806878 response to PegIFN-α treatment Increase in expression after IFN-γ treatment	Inhibit HBc degradation by UPS to positively affect replication
	TRIM11	mRNA transcripts increase linearly compared to viral load.	Unclear
HCV	TRIM59	High expression in NAFLD tissue Promote steatosis and ferroptosis	Interact and degrade GPX4 through ubiquitinated proteasome Enhance ubiquitination and degradation of p53
MAFLD/MASH	TRIM8	Promote NASH progression through JNK and NF-κB pathways Overexpression leads to hepatic steatosis	Inactivate TAK1 by ubiquitination, which can be blocked by TNIP3
	TRIM67	Enhance liver inflammation to disrupt lipid homeostasis and accumulate liver lipids Promote the progression of obesity-induced NAFLD	Induced by PGC-1α and HFD Increase extraction in FFA and reduce its oxidation
	TRIM15	Promote the proliferation and migration of HSCs	Positively regulate αSMA and collagen I Act on FAK in HSC activation
Liver fibrosis	TRIM52	Fibrosis promoter in vitro	Ubiquitinate PPM1A to mediate TGF-β/Smad signaling Downregulate PPM1A and induce dephosphorylated Smas2/3
	TRIM37	Promote liver fibrosis	Interact with SMAD7 and promote its ubiquitination-mediated degradation Downstream molecule of ROS
	TRIM23	Promote HSC Inhibit cell viability and activation	Overexpression reduces ferroptosis Enhance p53 ubiquitination
	TRIM47	Decrease inhibition of CTLD on fibrosis	Unclear

**Table 2 biomolecules-14-01038-t002:** The function and mechanism of the TRIM protein attenuating the progression of CLDs.

Diseases	TRIM Members	Functions	Molecular Mechanisms
HBV	TRIM25	Enhanced by HBx Promote HCC progression	Enhance GRP78 stability to promote MAN1B1 by HBx Activate PI3K/mTOR signaling pathway Induced by IFN-I in an IL-27-dependent manner
	TRIM21	Promote HBV DNA polymerase degradation Destroy stability of HBV DNA Inhibit DNA replication	Interact with TP domain using SPRY domain and degrade it
	TRIM14	Key molecule in IFN signal transduction pathway Prevent HBV replication	Induced by STAT 1 and activated by IFN-I Interact with HBx to inhibit the generation of Smc-HBx-DDB1 complex
	TRIM5γ	Inhibit HBV replication High expression level in IFN-α treatment indicates a better prognosis	Recruit TRIM31 and form a complex with HBx Promote the degradation of HBx protein through BBox domain
	TRIM22	Increase in response to IFN-γ treatment Help in HBV clearance	Inhibit the activity of HBV CP using SPRY domain
	TRIM56	Inhibit HBV replication	Degrade IκBα and phosphorylate p65 to impair HBV CP activity during cccDNA transcription
HCV	TRIM14	Effectively inhibit HCV replication and infection	Interact with NS5A to induce the degradation of the NS5A1 subdomain by the SPRY domain Induced by IFN-α and effectively inhibits HCV replication
	TRIM22	Upregulate in PBMCs of HCV patients after IFN treatment Participate in the antiviral effect induced by IFN-α	Interact with NS5A 1 domain and degrade NS5A in particular
	TRIM26	Host factors for HCV replication Contribute to host tropism	Interact with NA5B using the SPRY domain Mediate ubiquitination of NS5B Promote NS5B interaction with NS5A using RING domain
MAFLD/MASH	TRIM24	Inhibit spontaneous hepatic lipid production and accumulation in mice	Unclear
	TRIM28	Inhibit liver lipid accumulation	Recruited by SNX8 Mediate SNX8-FASN binding and ubiquitinate FASN
	TRIM16	Inhibit NASH progression Upregulate lipotoxic reactions Reduce liver steatosis, inflammation, and fibrosis induced by HFD and HFHC	Promote the degradation of phosphorylated TAK1 Weaken MAPK and JNK-p38 signaling pathway
	TRIM26	Prevent the progression of steatohepatitis Maintain liver homeostasis Regulate glucose and lipid metabolism	Inhibit CEPBD-HIF1A signaling Promote the ubiquitination of CEPBD
	TRIM31	Protective effect against hepatic steatosis	Bind Rhbdf2 and promote its proteasome degradation to inhibit Rhbdf2-MAP3K7 signaling
	TRIM38	Relieve the process of NAFLD/NASH	Attenuate the activation of the MAPK signaling pathway Promote TAB2 degradation
	TRIM65	Inhibit inflammatory response	Negatively regulate NLRP3 inflammasome Inhibit the macrophage polarization of the M1 type to M2 via JAK1/STAT1 signaling pathway
Liver fibrosis	TRIM26	Inhibit hepatic stellate cell activation to alleviate liver fibrosis	Mediate SLC7A11 ubiquitination Induce ferroapoptosis Promote lipid peroxidation and increase ROS levels Downregulate the GSH and NAPDH level
	TRIM33	Upregulate the response to hEMSC Alleviate liver fibrosis	Negative regulator of the TGF-β pathway Interact with SMAD2/3 family members as transcriptional corepressors Enhanced by HGF through 133 pCREBs
	TRIM24	Silencing disrupts liver homeostasis and develops liver fibrosis	Unclear

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
