# Peer review of "The Role of Tripartite Motif Family Proteins in Chronic Liver Diseases: Molecular Mechanisms and Therapeutic Potential"

_biomolecules, 2024, doi:10.3390/biom14081038_

Round 1

Reviewer 1 Report

Comments and Suggestions for Authors

This is an interesting review, worthy of publication. However, I would recommend the authors to consider the current terminology of NAFLD.

Comments on the Quality of English Language

Minor editing of English language required

Author Response

Comments 1: This is an interesting review, worthy of publication. However, I would recommend the authors to consider the current terminology of NAFLD.

Response 1: Thank you very much for your suggestion! We have updated the terms for NAFLD and NASH to the more generic terms "MAFLD" and "MASH." At the same time, the background and relevant information of this title have been pointed out in the article.

Reviewer 2 Report

Comments and Suggestions for Authors

Chronic liver disorders constitute a well known alarming problem worldwide due to their commonness and possible life threatening consequences. Therefore looking for  novel markers of their severity or target point for their treatment are of crucial importance. Authors of the manuscript performed the research of literature on the involvement of TRIM in the pathogenesis of liver diseases and their treatment. The most common hepatic entities are described in this context. This summary is comprehensive and detailed. Nonetheless, I would expand the list of references with other recent publications, e.g.: doi: 10.3389/fendo.2023.1210330, doi: 10.55730/1300-0144.5668. Furthermore, in the section of introduction I would suggest Authors to modify the section of introduction with additional information concerning other potential markers in the field of hepatology (except TRIM) to present a general future directions in the diagnostics of liver disorders.

Author Response

Comments: I would expand the list of references with other recent publications, e.g.: doi: 10.3389/fendo.2023.1210330, doi: 10.55730/1300-0144.5668. Furthermore, in the section of introduction I would suggest Authors to modify the section of introduction with additional information concerning other potential markers in the field of hepatology (except TRIM) to present a general future direction in the diagnostics of liver disorders.

Response: We appreciate your valuable suggestion! According to your suggestion, we updated the citations and added the following research articles to make the content more complete:

1) Zhang, Y.; Yuan, L.; Cui, S.; Wu, S. Tripartite motif protein 6 promotes hepatocellular carcinoma progression via multiple pathways. Turkish Journal of Medical Sciences 2023, 53, 1032 - 1044.

2) Vali, Y.; Lee, J.; Boursier, J.; Petta, S.; Wonders, K.; Tiniakos, D.; Bedossa, P.; Geier, A.; Francque, S.; Allison, M.; et al. Biomarkers for staging fibrosis and non-alcoholic steatohepatitis in non-alcoholic fatty liver disease (the LITMUS project): a comparative diagnostic accuracy study. The Lancet Gastroenterology & Hepatology 2023, 8, 714-725, doi:https://doi.org/10.1016/S2468-1253(23)00017-1.

3) Zhang, J.; Zhang, Y.; Ren, Z.; Yan, D.; Li, G. The role of TRIM family in metabolic associated fatty liver disease. Front Endocrinol (Lausanne) 2023, 14, 1210330, doi:10.3389/fendo.2023.1210330.

4) Hu, X.; Tang, Z.; Ma, S.; Yu, Y.; Chen, X.; Zang, G. Tripartite motif-containing protein 7 regulates hepatocellular carcinoma cell proliferation via the DUSP6/p38 pathway. Biochem Biophys Res Commun 2019, 511, 889-895, doi: 10.1016/j.bbrc.2019.02.001.

At the same time, according to your thoughtful suggestions, we searched the literature on biomarkers related to liver disease, focusing on their application in diagnosis. We have added the following to the article:

“As a potential diagnostic biomarker, the detection of TRIM protein is of great significance, in addition to helping to evaluate the progression and prognosis of liver disease, it can also use to detect drug therapy effect and advance preventive screening [13]. Except for TRIM proteins, other markers such as ADAPT and MACK-3 have also been shown to evaluate the progression of liver fibrosis and MASH. This will also improve the efficiency of screening people for clinical drug trials, avoiding invasive tests such as liver biopsies [14].” (Page 2)

Reviewer 3 Report

Comments and Suggestions for Authors

I consider the manuscript by Xiwen Cao and collaborators to be interesting and well-written. Some suggestions

1. Table 1 is very large. Can you divide it into two or three tables?

2. The work is very well-written, but the conclusions are short. Can you expand on them?

3. Can you add perspectives?

4. The figures in the manuscript are of very good quality. Can you add more figures in the different sections? For example, put a figure in the conclusions of the project proposal?

Thank you

Author Response

Comments 1: Table 1 is very large. Can you divide it into two or three tables?

Response 1: Thank you very much, your suggestions are very helpful! According to your suggestion, we divided Table 1 into two charts according to the effect of promoting or attenuating the progress of CLDs. They were named " Table 1: The function and mechanism of TRIM protein promoting the progression of CLDs" and " Table 2: The function and mechanism of TRIM protein attenuating the progression of CLDs " (pp. 10-14)

Comments 2: The work is very well-written, but the conclusions are short. Can you expand on them?

Response 2: According to your comments, we have expanded the conclusion part again. Firstly, we thought more deeply about the relationship between MAFLD and its associated diseases. “As a systemic-effect metabolic disease, MAFLD is closely associated with IR and extrahepatic metabolic diseases. Hepatic insulin resistance has been found to be regulated by TRIM protein, and we can speculate that the expression of TRIM protein is widespread and may play an important role in the metabolic pathways of various organs [97].” (Page 17)

Secondly, we discussed the association between CLDs diseases in more depth: “At the same time, we also found that the progression of CLDs is continuous, and different liver diseases can be transformed into each other, patients with hepatitis B and C are prone to HCC, and liver fibrosis is the progressive form of MAFLD/MASH. TRIM proteins play an important regulatory role in these diseases. Previous studies have shown that the expression level of TRIM proteins changes significantly during the progression of HBV infection to HCC, and TRIM proteins also regulate the lipid peroxidation mechanism involved in both MAFLD and liver fibrosis [98]. This strong correlation can provide us with new ideas for exploring targets.” (Page 17-18)

Comments 3: Can you add perspectives?

Response 3: In addition to existing content, we discuss the universality and diversity of TRIM protein, and prove it with specific examples in this paper to provide new ideas for target drugs. “The expression of TRIM proteins is universal and diverse. Even from the same subfamily with similar structure, it may also have uncertain effects. Some can play the same role in the face of the same disease, such as TRIM28 and TRIM38 in the progression of MAFLD, and TRIM15 and TRIM47 in liver fibrosis. There are also opposite regulatory effects, such as TRIM21 and TRIM26 in HBV infection, and TRIM11, TRIM22, and TRIM26 in HCV infection. Surprisingly, the same molecule can play multiple roles in the same disease, such as TRIM25, which inhibits viral replication while also promoting HCC progression. This means that in order to ensure curative effect, the future treatment of TRIM needs to rely on precision medicine therapy to accurately target drug delivery.” (Page 18)

At the same time, according to your thoughtful suggestions, we searched the literature on biomarkers related to liver disease, focusing on their application in diagnosis. We have added the following to the article:

“As a potential diagnostic biomarker, the detection of TRIM protein is of great significance, in addition to helping to evaluate the progression and prognosis of liver disease, it can also use to detect drug therapy effect and advance preventive screening [13]. Except for TRIM proteins, other markers such as ADAPT and MACK-3 have also been shown to evaluate the progression of liver fibrosis and MASH. This will also improve the efficiency of screening people for clinical drug trials, avoiding invasive tests such as liver biopsies [14].” (Page 2)

Comments 4: The figures in the manuscript are of very good quality. Can you add more figures in the different sections? For example, put a figure in the conclusions of the project proposal?

Response 4: This is really helpful advice! According to your suggestions, we decided to add a figure in the conclusion to briefly reflect the mechanism between TRIM protein and CLDs, and make a summary of the previous discussion.

The figure is as follows:

Reviewer 4 Report

Comments and Suggestions for Authors

Comments for paper

 Comments to the Authors:

In this work, Cao and colleagues presented an exciting review on the molecular mechanisms and therapeutic potential of TRIM proteins in chronic liver diseases.

Below, I have outlined some points for the authors to address:

1.   Recently, the nomenclature of NAFLD and NASH have been changed to MAFLD and MASH. This new nomenclature should be used throughout the article.

2.  The authors should mention the methods or approaches used to search articles related to the role of TRIM proteins in chronic liver diseases vis-a-vis their molecular mechanisms and therapeutic potential in the introduction.

 3.   Although in this review, the authors are focusing on the role of TRIM  proteins in the pathobiology of chronic liver diseases arising from Chronic Viral Hepatitis and MAFLD/MASH,  it will appeal to a wider readership if they could mention their roles in the pathogenesis of  chronic liver diseases consequent of other etiologies such as Alcoholic Liver Disease and others. Addition of such account will make the article more robust and give a broader understanding of role of TRIM proteins in chronic liver diseases pathology and aid future research.

3)   For clarity, Table1 should be separated into two. With one table showing the TRIM family members that are promoting the progression of chronic liver diseases and the other showing those TRIM members that are attenuating or regressing the progression of chronic liver diseases

4)  The authors should  cross-check the list of abbreviations, as some of the abbreviations used in the main text might not have been listed.

5) There is a need for minor editing and grammar checks in the manuscript.

Comments on the Quality of English Language

Minor editing of English language and grammar checks are neccessary in the mauscript

Author Response

Comment 1.   Recently, the nomenclature of NAFLD and NASH have been changed to MAFLD and MASH. This new nomenclature should be used throughout the article.

Response 1: Thank you very much for your suggestion! We have updated the terms for NAFLD and NASH to the more generic terms "MAFLD" and "MASH." At the same time, the background and relevant information of this title have been pointed out in the article.

Comment 2. The authors should mention the methods or approaches used to search articles related to the role of TRIM proteins in chronic liver diseases vis-a-vis their molecular mechanisms and therapeutic potential in the introduction.

Response 2: According to your suggestion, we have added the missing information in the article as follows: “In order to comprehensively review and analyze the advanced mechanism of TRIM proteins in CLDs, a systematic literature search strategy was adopted in this review. We mainly focused on relevant research results in the past few years. Searches are made in multiple academic databases, including PubMed, Web of Science, Scopus, and Google Scholar. The search terms included "TRIM protein", "HBV ","HCV", "MAFLD (NAFLD)","MASH(NASH)", "chronic liver disease", which were combined with the study type and language category. Through this process, we aim to collect and analyze the most representative and influential literature within the field to provide a comprehensive perspective.” (pp.1-2)

Comment 3. Although in this review, the authors are focusing on the role of TRIM proteins in the pathobiology of chronic liver diseases arising from Chronic Viral Hepatitis and MAFLD/MASH, it will appeal to a wider readership if they could mention their roles in the pathogenesis of chronic liver diseases consequent of other etiologies such as Alcoholic Liver Disease and others. Addition of such account will make the article more robust and give a broader understanding of role of TRIM proteins in chronic liver diseases pathology and aid future research.

Response 3: Thank you very much for your comments, which is exactly what we were hoping to do in the beginning - look for more chronic diseases associated with TRIM to make the review broader. Unfortunately, in various library searches, most of the researchers focused on chronic viral hepatitis, MAFLD (NAFLD), and no relevant studies have reported the relationship between alcoholic fatty liver and TRIM, and the rest of the literature is not very closely related to our topic. Therefore, it is a pity that we did not find other useful articles to enrich the content of the review. We are very sorry that we did not meet your requirements. Hope you can understand.

Comment 4. For clarity, Table1 should be separated into two. With one table showing the TRIM family members that are promoting the progression of chronic liver diseases and the other showing those TRIM members that are attenuating or regressing the progression of chronic liver diseases.

Response 4: Thank you very much, your comments are great! It is very helpful to our review. According to your suggestion, we divided Table 1 into two charts according to the effect of promoting or attenuating the progress of CLDs. (pp. 10-14)

Comment 5. The authors should cross-check the list of abbreviations, as some of the abbreviations used in the main text might not have been listed.

Response 5: Thank you very much for your suggestion, please forgive our carelessness, we have re-checked the completeness of the abbreviation list to make sure that it matches the article exactly.

Comment 6. There is a need for minor editing and grammar checks in the manuscript.

Response 6: Please be assured that we have re-checked the text word for word to ensure that there is no improper use of grammar and vocabulary. Thank you very much for your comments, which have greatly helped to improve the quality of our manuscripts!

Reviewer 5 Report

Comments and Suggestions for Authors

The authors summarized many of the latest research that tripartite motif family proteins (TRIMs) function as E3 ligases, which participate in the progression of chronic liver diseases (CLDs) by regulating gene and protein expression levels through post-translational modification. They try to clarify the molecular mechanisms and potential therapeutic targets of TRIMs in CLDs and provide insights for therapy guidelines and future research. This reviewer has some comments.

Comments:

1.      The characteristics of TRIM in each etiology of CLD are described here, but it is unclear whether the expression and function of TRIM are similar or different in each etiology. It is important as a therapeutic target.

2.      Regarding the TRIM in HBV and HCV, it is unclear whether the expression level and function of TRIM are affected by the amount and type of virus.

Author Response

Comment 1. The characteristics of TRIM in each etiology of CLD are described here, but it is unclear whether the expression and function of TRIM are similar or different in each etiology. It is important as a therapeutic target.

Response 1: Thank you very much for your suggestion, which is very helpful to improve the quality of our manuscript. According to your suggestions, we have re-sorted out the regulation of different TRIM proteins in diseases, and in the conclusion, we have supplemented the diversity of their functions and summarized some findings of existing studies. The supplementary content is as follows: “The expression of TRIM proteins is universal and diverse. Even from the same subfamily with similar structure, it may also have uncertain effects. Some can play the same role in the face of the same disease, such as TRIM28 and TRIM38 in the progression of MAFLD, and TRIM15 and TRIM47 in liver fibrosis. There are also opposite regulatory effects, such as TRIM21 and TRIM26 in HBV infection, and TRIM11, TRIM22, and TRIM26 in HCV infection. Surprisingly, the same molecule can play multiple roles in the same disease, such as TRIM25, which inhibits viral replication while also promoting HCC progression. This means that in order to ensure curative effect, the future treatment of TRIM needs to rely on precision medicine therapy to accurately target drug delivery.” (Page 18) We also supplemented the role of TRIM protein in related diseases: “At the same time, we also found that the progression of CLDs is continuous, and different liver diseases can be transformed into each other, patients with hepatitis B and C are prone to HCC, and liver fibrosis is the progressive form of MAFLD/MASH. TRIM proteins play an important regulatory role in these diseases. Previous studies have shown that the expression level of TRIM proteins changes significantly during the progression of HBV infection to HCC, and TRIM proteins also regulate the lipid peroxidation mechanism involved in both MAFLD and liver fibrosis [98]. This strong correlation can provide us with new ideas for exploring targets.” (pp. 17-18) It can be seen that there are many mechanisms to be discovered in this point, which need to be further explored through research.

Comment 2. Regarding the TRIM in HBV and HCV, it is unclear whether the expression level and function of TRIM are affected by the amount and type of virus.

Response 2: Thank you very much for your suggestions, which gave us a lot of inspiration and thinking. As a post-translational modification protein, TRIM plays a regulatory role in molecular mechanisms. Many literatures we reviewed focused on its regulation of physiological activities such as viral replication without focusing on the internal mechanism of virus's influence on it. However, according to existing studies, they are mutually influenced, and the expression level of TRIM protein will change with the change of the amount of virus. We have added this thinking to the manuscript: “While the TRIM proteins play a role in regulating viral replication, the viral load also affects the expression level of the protein. For example, TRIM22 upregulation is associated with HCV decline during IFN-α treatment. At the same time, in some TRIM proteins that play the function of ISG (TRIM14, TRIM25, etc.), the level of virus indirectly promotes the expression of TRIM proteins [55]. (Page 16)